# Personalized Model to Predict Small for Gestational Age at Delivery Using Fetal Biometrics, Maternal Characteristics, and Pregnancy Biomarkers: A Retrospective Cohort Study of Births Assisted at a Spanish Hospital

**DOI:** 10.3390/jpm12050762

**Published:** 2022-05-08

**Authors:** Peña Dieste-Pérez, Ricardo Savirón-Cornudella, Mauricio Tajada-Duaso, Faustino R. Pérez-López, Sergio Castán-Mateo, Gerardo Sanz, Luis Mariano Esteban

**Affiliations:** 1Department of Obstetrics and Gynecology, Miguel Servet University Hospital and Aragón Health Research Institute, 50009 Zaragoza, Spain; mtajadad@gmail.com (M.T.-D.); scastan@salud.aragon.es (S.C.-M.); 2Department of Obstetrics and Gynecology, San Carlos Clinical Hospital and San Carlos Health Research Institute (IdISSC), Complutense University, 28040 Madrid, Spain; rsaviron@gmail.com; 3Department of Obstetrics and Gynecology, University of Zaragoza Faculty of Medicine and Aragón Health Research Institute, 50009 Zaragoza, Spain; faustino.perez@unizar.es; 4Department of Statistical Methods and Institute for Biocomputation and Physics of Complex Systems-BIFI, University of Zaragoza,50018 Zaragoza, Spain; gerardo@unizar.es; 5Engineering School of La Almunia, University of Zaragoza, 50100 La Almunia de Doña Godina, Spain

**Keywords:** small for gestational age, estimated percentile weight, combined prediction model, fetal biometry, third trimester ultrasound, pregnancy biomarkers

## Abstract

Small for gestational age (SGA) is defined as a newborn with a birth weight for gestational age < 10th percentile. Routine third-trimester ultrasound screening for fetal growth assessment has detection rates (DR) from 50 to 80%. For this reason, the addition of other markers is being studied, such as maternal characteristics, biochemical values, and biophysical models, in order to create personalized combinations that can increase the predictive capacity of the ultrasound. With this purpose, this retrospective cohort study of 12,912 cases aims to compare the potential value of third-trimester screening, based on estimated weight percentile (EPW), by universal ultrasound at 35–37 weeks of gestation, with a combined model integrating maternal characteristics and biochemical markers (PAPP-A and β-HCG) for the prediction of SGA newborns. We observed that DR improved from 58.9% with the EW alone to 63.5% with the predictive model. Moreover, the AUC for the multivariate model was 0.882 (0.873–0.891 95% C.I.), showing a statistically significant difference with EPW alone (AUC 0.864 (95% C.I.: 0.854–0.873)). Although the improvements were modest, contingent detection models appear to be more sensitive than third-trimester ultrasound alone at predicting SGA at delivery.

## 1. Introduction

Fetal growth restriction (FGR) is defined as a failure to achieve the endorsed growth potential. This definition includes the so-called true FGR, which associates alterations in the Doppler study, suggesting a hemodynamic redistribution that reflects fetal adaptation to malnutrition/hypoxia, as well as histological and biochemical signs of placental disease with an increased risk of preeclampsia [1]. These fetuses have a 5- to 10-fold increased risk of death in utero and increased risk of perinatal morbidity and mortality and suboptimal long-term outcomes [2,3,4,5]. This group also includes fetuses who were referred to as small for gestational age, whose estimated fetal weight (EFW) was below a certain threshold, most commonly the 10th percentile [6,7]. They also have a higher morbidity and perinatal mortality but are not usually associated with the Doppler signs described for FGR. Finally, a subgroup of the above corresponds to so-called “constitutionally small” fetuses, which are born small, with an estimated percentile weight (EPW) below the 10th percentile, but are otherwise healthy [8].

While these definitions seem conceptually simple, the distinction in clinical practice can be challenging. Most small for gestational age (SGA) babies go unnoticed until birth, even when a routine third-trimester ultrasound is performed [9,10]. On the other hand, this category misses cases of growth restriction that do not fall below the 10th percentile. In spite of this, this definition can still help to identify a subset of pregnancies considered as high risk [1].

Nowadays, the diagnostic strategy for the detection of these fetuses prenatally is routine third-trimester ultrasound, performed around 35–37 weeks of gestation, which evaluates fetal growth. However, this has quite low detection rates (DR), ranging from 50% to 80% [11], and the impact of this on perinatal outcome is unclear [12].

For this reason, the addition of other markers, such as maternal characteristics and biochemical and biophysical parameters, is being studied. Hence, combined models are being designed that either increase the predictive capacity of basic ultrasound in the third trimester of pregnancy to predict SGA or select patients at risk of giving birth to late-onset SGA fetuses [12,13,14,15,16,17,18]. In some of these studies, an ultrasound is performed well before delivery (week 30–34) [12,15,19]; in others, several ultrasounds are performed throughout the third trimester of pregnancy, in order to longitudinally assess fetal growth [20]. In others, the Doppler study or circulating biochemical markers, such as serum placental growth factor (PlGF) and soluble fms-like tyrosine kinase-1 (sFLT), are introduced, thus increasing the sensitivity and specificity, as well as the detection rates, of SGA fetuses. However, the above strategies are not routinely performed in low-risk pregnancies [5,13,18,19,21,22,23,24].

Recent evidence suggests that the pathologies underlying FGR and SGA take place in the first trimester. An earlier assessment, before the establishment of placental dysfunction, may have the potential to improve treatment and prognosis in clinical practice [25]. The cost effectiveness would be even greater if this identification could be a spinoff from the widely-implemented first trimester combined ultrasound and biochemical screening program for Down’s syndrome, which tests for maternal serological markers pregnancy-associated plasma protein A (PAPP-A) and the beta subunit of human chorionic gonadotrophin (β-hCG) [26].

Some studies have already evaluated the individual capacity of PAPP-A and β-hCG to predict SGA. They found that these markers have an independent influence on the final birth weight and correlated a lower PAPP-A with a higher risk of the fetus developing SGA. However, their predictive powers are insufficient for them to be used alone for SGA detection [27,28,29,30].

The objective of our study was to compare the predictive capacity for SGA neonates of fetal biometry, performed in the third-trimester ultrasound on all pregnant women in a Spanish hospital between 35 and 37 weeks of gestation, with a multivariate model composed of the aforementioned ultrasound, plus maternal characteristics and biochemical markers used for the screening of chromosomal abnormalities in the first trimester of gestation (PAPP-A and β-HCG), tests which are performed in all low-risk pregnant women.

## 2. Material and Methods

### 2.1. Study Design

This was a retrospective cohort study of births assisted at the Miguel Servet University Hospital (MSUH), Zaragoza, Spain, between March 2012 and December 2018. The inclusion criteria were as follows: live singleton pregnancies, controlled at the MSUH from the first trimester of gestation; fetal ultrasound assessment at a gestational age of 35 weeks (range 34–36); and deliveries between 37 and 42 weeks of gestational age, with fetuses without stillbirth associated with malformations or chromosomal abnormalities. Of the 16,361 deliveries assisted in our hospital in the study period, only the 12,912 cases that fulfilled the inclusion criteria, such as data availability to estimate percentile weights by standards, were considered. The selected sample of study participants is described, in detail, in Figure 1.

The last menstrual period was adjusted by the first trimester ultrasound [31]. Universal ultrasound screening was performed at 35 weeks (range 34–36 weeks) at the Ultrasound and Prenatal Diagnosis Unit, using either an ultrasound machine Voluson 730 Expert, E6, E8 (General Electric, Healthcare, Zipf, Austria) or Aloka Prosound SSD-5000 (Hitachi Aloka Medical Systems, Tokyo, Japan). This ultrasound test is routinely performed in all pregnancies at our center, in an attempt to increase the detection of fetal growth alterations. EFW was calculated with the formula of Hadlock et al. [32], which combines biparietal diameter, cephalic and abdominal circumference (AC), and femur length. 

### 2.2. Estimated Percentile Weight

EPWs were calculated according to the local MSUH standard, customized to fetal gender, built using a modified version of Hadlock et al. growth charts [33], and adjusted to our population, with a coefficient of variation that changes with gestational age [34]. To assess ultrasound weight measures in the third trimester, EPWs were estimated between 34 and 36 weeks of gestational age. As a gold standard for the analysis, SGA was defined as percentile birth weight under 10, using a growth reference for the Spanish population, based on 9362 birth weights [35].

### 2.3. Estimated Abdominal Circumference Percentile 

The AC percentile was estimated according to the Smulian et al. methodology [36]. These authors have derived a formula, based on 10,070 fetuses, for calculating the mean and standard deviation, depending on gestational age. Then, assuming a normal distribution for AC measures at a gestational age, the percentiles were estimated.

### 2.4. Statistical Analysis

Data were descriptively analyzed using the medians and interquartile ranges for continuous variables and absolute and relative frequencies for categorical variables. Differences between SGA and non-SGA groups were tested using Mann–Whitney and chi-square tests, as appropriate.

The predictive ability of EPW, provided by the MSUH standard, to predict SGAs was analyzed using the area under the receiver operating characteristic curve (AUC) [37]. This area is equivalent to the probability that, given two individuals, one SGA and the other non-SGA, the marker assigns a greater probability of being SGA to the individual that is really SGA. The area ranges from 0.5 to 1, with the 0.5 value corresponding to a random model, 0.7 to an acceptable model, 0.8 to a good model, 0.9 excellent model, and 1 to perfect discrimination.

To improve the prediction of SGA, we explored the added predictive ability of maternal–fetal characteristics and pathologies. These corresponded to: maternal age and body mass index at the start of pregnancy, maternal height, parity, previous cesarean, in vitro fertilization, infant gender, PAPP-A, β-HCG, smoking habits, hypertension, and diabetes. In addition, the AC percentile, estimated at the 35th week of gestational age, was added as a complementary predictor of the EPW to identify SGA fetuses at birth.

AUCs were compared using a bootstrap test [38], and the best model was taken as the one with the largest AUC value. Calibration and clinical utility analysis, by means of a calibration curve [39] and clinical utility curve (CUC) [40], complemented the validation process of the predictive model derived.

Calibration graphically analyzes the concordance between the predictions and real occurrence of the outcome, usually through calibration curves and two informative parameters: ‘intercept’ (calibration-in-the-large), which measures the difference between average predictions and average outcome; and ‘slope’, which reflects the average effect of predictions on the outcome.

The CUC reflects the consequences of choosing a cutoff point, in terms of patients with a wrongly classified outcome of interest versus processes avoided. In this curve, the X axis corresponds to the possible threshold probability points, and the Y axis represents the percentage of two measures; the first corresponds to the percentage of missing positive cases below the selected cut-off (FN), and the second to the number of individuals below the cut-off.

Analyses were performed using the R version 4.0.3 language programming package (The R Foundation for Statistical Computing, Vienna, Austria) [41]. 

## 3. Results

### 3.1. Descriptive Results

Table 1 shows the descriptive characteristics of the pregnancies for the SGA and non-SGA groups. For the standard calculation of EFWs, by ultrasound alone at 35 weeks (range 34 + 0 to 36 + 6 weeks), an EPW value < 10 detects 42.1% SGA at birth. The remaining 57.9% correspond to EPW > 10 at the 35th week of gestational age. An AC percentile of <10 at 35 weeks detects only 18.5% of SGA at birth. The variables body mass index, maternal height, parity, number of previous cesareans, in vitro fertilization, maternal smoking habits, hypertension, PAPP-A, and β-HCG all showed statistically significant differences between SGA and non-SGA groups. 

### 3.2. Small for Gestational Age Prediction

We explored EPW as a predictor of SGA using a logistic regression model, with EPW adjusted for restricted cubic splines with four knots. The AUC was 0.864 (0.854–0.873 95% C.I.), showing a good discriminative ability.

Moreover, we constructed a multivariate model by adding maternal–fetal characteristics and AC percentiles. Table 2 shows the hazard ratio, 95% CI, and *p*-values for significant variables.

The AUC for the multivariate model was 0.882 (0.873–0.891 95% C.I.), showing a statistically significant difference with the EPW at week 35 (*p*-value < 0.001), although the increase in AUC was modest. The ROC curves for both models are presented in Figure 2. Regarding the added predictive ability of the AC percentile, a multivariate model without this variable had an AUC value of 0.880, with no significant difference from the full model (*p*-value = 0.067). Table 3 shows the added predictive ability of each marker, measured by AUC, and discrimination rate for a 10% false-positive rate (FPR).

For the validation process of the EPW at the 35th week and multivariate model to predict SGA, the calibration was explored (Figure 3). Both models showed a good agreement between the predicted probability and actual occurrence, with an intercept of 0 and a slope of 1, corresponding to a perfect calibration.

Finally, we analyzed the clinical utility. Figure 4 shows the CUC for EPW at the 35th week (top panel), as well as for the multivariate model (bottom panel).

The better performance of a marker is reflected in a greater separation of the curves plotted in the CUC. Using the EPW at the 35th week, for a 6% cutoff point in a logistic regression model, 11.5% of SGA would be wrongly classified, with 61.5% of fetuses at low risk of being SGA. For the same cutoff point, in the multivariate model, 10.9% of SGA would be wrongly classified with 63.9% of fetuses at low risk of being SGA. A slightly better performance was, therefore, obtained with the multivariate model. 

## 4. Discussion

Our findings show that a combined screening model, including EPW and AC percentile by ultrasound at the 35th week, maternal characteristics, and biochemical markers, had a better performance than EPW alone in predicting SGA. The combined model presented higher AUC than the model with only EPW. These differences were significant, but the increase was modest. The combined model, without AC percentile, did not show significant differences from the complete model. Moreover, with the combined screening model, 3% fewer fetuses required control for a high risk of SGA.

Our DR improved from 58.9% (threshold EPW 18.2%) using EPW, or from 52.3% with the AC percentile alone (threshold percentile AC 24.9%) to 63.5% using the predictive model, for a 10% FPR. However, this improvement is limited and comparable to the findings of other studies predicting neonates with a birth weight < 10th centile at, or after, term using combined models. These report DRs between 51% and 74%, at a 10% FPR [16,17,42,43], although the markers used in the predictive models are different. In addition, we used cut-off points higher than the 10th percentile, as a cut-off point of 10 was insufficient, in line with other publications [8,44].

The biochemical markers used in our study, β-hCG and PAPP-A, are routinely tested in the first trimester of pregnancy to screen for chromosomal disorders, and their correlations with chromosomal disorders are already known. Hence, PAPP-A is an independent factor influencing final birth weight, and the lower the PAPP-A, the higher the risk of a fetus developing SGA. However, their predictive powers are not sufficient for them to be used alone for SGA detection [27,28,29,30]. Moreover, a significant positive correlation has not been found between birth weight and free β-hCG levels [30]. These results are consistent with our findings of lower PAPP-A and β-hCG values in the group of SGA fetuses than in the group of non-SGA fetuses, both with significant differences.

With regards to combined SGA prediction models, a few studies have examined the performance of screening for SGA at 35–37 weeks’ gestation by combining EFW and different markers. One study of 5121 pregnancies reported that, in screening by maternal factors and EFW, the DR of SGA < 10th percentile delivering at >37 weeks was 66%, at a 10% screen-positive rate, and this did not improve with addition of the artery pulsatility index (UtA-PI) and mean arterial pressure [18]. Similarly, a study of 946 pregnancies reported that screening by EFW predicted 59% of SGA < 10th percentile, at a 10% screen- positive rate, and the performance was not improved, either by the addition of UtA-PI or the cerebroplacental ratio [45]. In yet another study of 3859 pregnancies, screened by maternal factors and EFW, the DR of SGA < 10th percentile delivering at >37 weeks was not improved by the addition of PlGF and sFLT [46]. 

On the other hand, Miranda et al. 2017 used a combined screening model, including a priori risk (maternal characteristics), third trimester (32 + 0 to 36 + 6) EPW, UtA-PI, PlGF, and estriol (with lipocalin-2 for SGA), and achieved a DR of 61% (AUC, 0.86 (95% confidence interval CI, 0.83–0.89)) for SGA cases and 77% (AUC, 0.92 (95% CI, 0.88–0.95)) for FGR. The combined model performed significantly better than using EPW alone (*p* < 0.001 and *p* = 0.002, respectively) [21]. Despite using different biomarkers and not adding Doppler ultrasound, our SGA DRs in the combined model were 63.5% (AUC 0.882 0.873–0.891 95% C.I.).

In their combined model, Souka et al. 2012 used AC, EFW, UA Doppler, smoking status, and first-trimester indices (free β-hCG and PAPP-A multiples of the median) and obtained an AUC = 0.88 for the prediction of SGA, a marginal improvement on EPW or AC alone, but without statistically significant differences [13]. The results of our work were very similar, without the use of AU Doppler, which is not routinely performed when the EFW is above the 10th percentile. 

Ciobanu et al. reported a positive DR of 32% (95% CI, 30–36%) in the detection by maternal factors, 66% (95% CI, 63–69%) by maternal factors and EFW at 35–36 weeks of gestation, and 69% (95% CI, 66–72%) with the addition of biomarkers (UtA-PI, umbilical artery pulsatility index, middle cerebral artery pulsatility index, PlGF, and sFLT) [5]. In our cohort, these values were 26.3% (95% CI 23.9–28.8%) with maternal factors alone, 62.1% (95% CI 59.4–64.7%) with the addition of EPW, and 62.4% (95% CI 59.7–64.0%) with the complete combined model. 

The strengths of our screening model are its simplicity and affordability, as it includes the standard tests used in screening for chromosomal abnormalities in the first trimester. It is based on variables easily obtained in the routine control of normal pregnancy, without requiring additional tests or parameters to elaborate the predictive model, such as Doppler studies or angiogenic biomarkers.

Several studies have shown that the performance of screening for SGA using a combined model of maternal characteristics and medical history (maternal factors), EFW, and biophysical and biochemical markers is acceptably high for a preterm birth, but disappointingly low for delivery at term [15,42]. Both in our study and in most of those cited here, the contribution of a model that combines maternal characteristics and medical history (maternal factors), EFW, and biophysical and biochemical markers increases the predictive capacity of SGA fetuses, but only to a small degree. However, other studies have shown this to be acceptably high for mothers who give birth prematurely [15,42].

We analyzed the clinical utility of EPW at the 35th week, as well as the predictive model using maternal–fetal characteristics, by means of the CUC. In this curve we showed the percentage of SGA incorrectly classified using a threshold point, as well as the fetuses at low risk of being SGA at birth. For the EPW at week 35, assuming a loss of 10% of fetuses that would be SGA at birth, 59% of fetuses can be considered as low risk. Alternatively, assuming a loss of 20% SGA cases, 71% of fetuses would be at low risk. Using the predictive model, assuming a loss in SGA cases of 10%, 62% would be considered as low risk; with a loss of 20%, a total of a 74% would be at low risk. From these findings, it can be deduced that, with the addition of maternal fetal characteristics, 3% fewer fetuses would require more controlled follow-up.

Detection of SGA at delivery by third-trimester ultrasound, either by EPW or CA, even with models combined with other maternal variables and first-trimester biochemical markers, is limited, and new tools are required to improve this.

## 5. Conclusions

Contingent screening models appear to be more sensitive than third-trimester ultrasound screening as the sole technique for predicting SGA at delivery. However, these improvements are modest (from 58.9% using EPW or 52.3% with AC percentile alone to 63.5% using the predictive model). AC at 35-week ultrasound does not appear to be superior to EPW or significantly improve on the full model. 

## Figures and Tables

**Figure 1 jpm-12-00762-f001:**
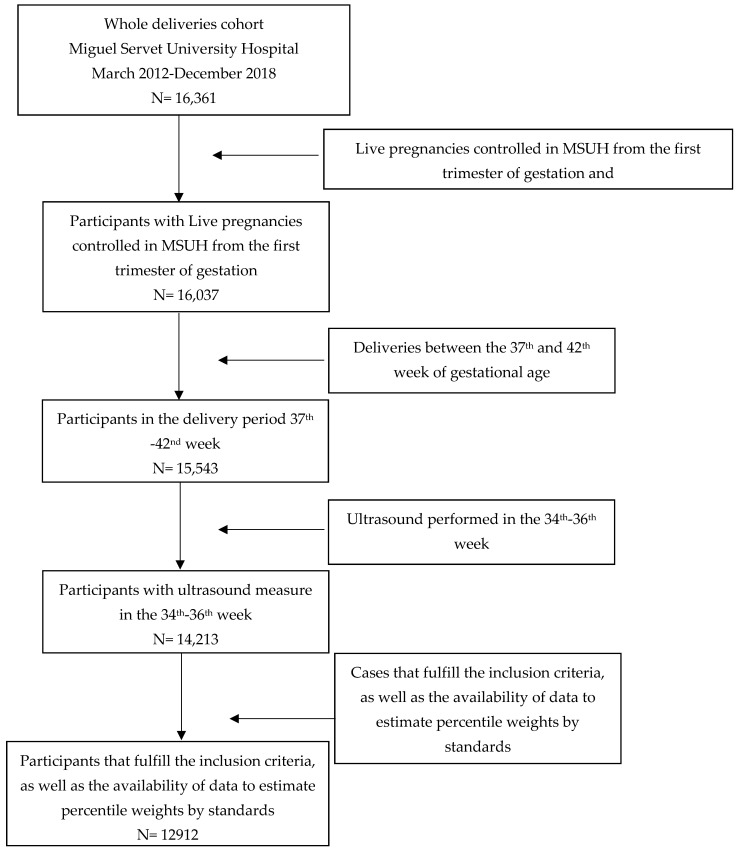
Study participants selection sample.

**Figure 2 jpm-12-00762-f002:**
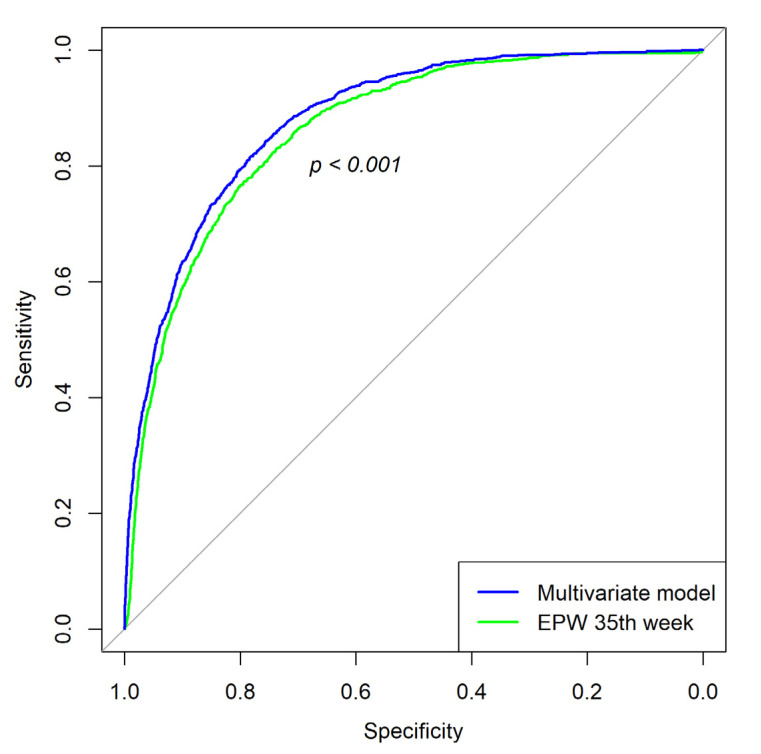
ROC curve for the EPW at 35th week and multivariate model.

**Figure 3 jpm-12-00762-f003:**
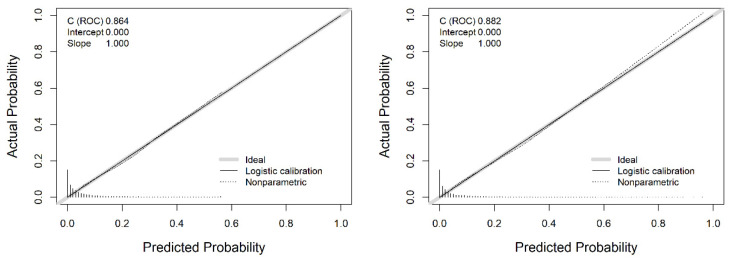
Calibration curve for the EPW at 35th week (left panel) and multivariate model (right panel).

**Figure 4 jpm-12-00762-f004:**
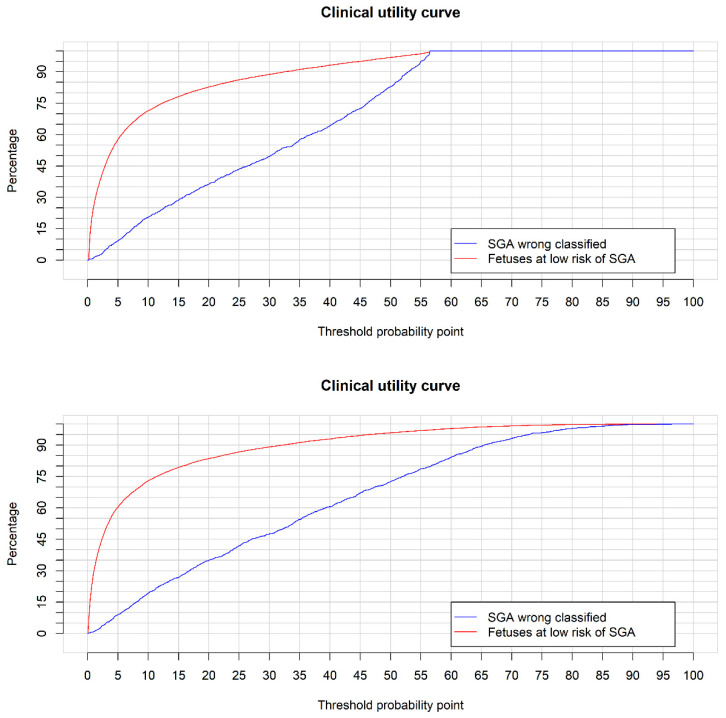
The CUC for EPW at 35th week (top panel) and multivariate model (bottom panel).

**Table 1 jpm-12-00762-t001:** Maternal baseline characteristics (top), pregnancy (middle), and perinatal characteristics (bottom) of pregnancies. Data are reported as *n* (%) or medians (interquartile range). MSUH, Miguel Servet University Hospital.

Clinical Characteristics	Pregnancies SGA (*n* = 1281)	Pregnancies Non-SGA (*n* = 11,631)	*p*-Value
Maternal characteristics			
Maternal age (years)	33.4 (29.9–36.4)	33.2 (30.0–36.1)	0.299
Maternal body mass index (kg/m^2^)	22.5 (20.7–25.4)	23.4 (21.2–26.4)	<0.001
Maternal height (cm)	161 (157–165)	163 (160–168)	<0.001
Parity			
0	872 (68.1%)	6151 (52.9%)	<0.001
1	339 (26.5%)	4411 (37.9%)	
≥2	70 (5.4%)	1069 (9.2%)	
Previous cesarean			
0	1211 (94.5%)	10,739 (92.3%)	0.004
1	69 (5.4%)	826 (7.1%)	
≥2	1 (0.1%)	66 (0.6%)	
In vitro fertilization			
No	1217 (95.0)	11,121 (95.6%)	0.394
Yes	64 (5.0%)	510 (4.4%)	
Maternal smoking habits			
Yes	352 (27.5%)	1676 (14.4%)	<0.001
No	929 (72.5%)	9955 (85.6%)	
Hypertension			
No	1235 (96.4%)	11,485 (98.7%)	<0.001
Chronic	5 (0.4%)	25 (0.2%)	
Preeclampsia	18 (1.4%)	47 (0.4%)	
Gestational	23 (1.8%)	74 (0.6%)	
Diabetes			
No	1126 (87.9%)	10,356 (89.0%)	0.343
Pregestational	6 (0.5%)	81 (0.7%)	
Gestational	132 (10.3%)	1043 (9.0%)	
Carbohydrate intolerance	17 (1.3%)	151 (1.3%)	
Ultrasound parameters at 35 (34–36) weeks			
Gestational age (weeks) at ultrasound	35.1 (35.0–35.3)	35.1 (35.0–35.3)	0.345
Estimated fetal weight (grams) by Hadlock	2186 (2042–2349)	2532 (2362–2715)	<0.001
Abdominal fetal circumference (cm)	293 (284–301)	311 (302–321)	<0.001
Percentile weight by MSUH standard			
<10	513 (42.1%)	542 (4.7%)	<0.001
≥10	768 (57.9%)	11,089 (95.3%)	
Percentile AC by Smulian standard			
<10	237 (18.5%)	200 (17.6%)	<0.001
≥10	1044 (81.5%)	11,431 (82.4%)	
Pregnancy and perinatal outcomes			
PAPP-A	0.84 (0.57–1.25)	0.99 (0.68–1.42)	<0.001
β-HCG	0.91 (0.61–1.42)	1.00 (0.67–1.51)	<0.001
Gestational age at delivery	39.6 (38.7–40.4)	40.7 (40.0–41.3)	<0.001
Newborn gender			
Female	663 (51.8%)	5617 (48.3%)	0.020
Male	618 (48.2%)	6014 (51.7%)	
Birth weight	2650 (2480–2760)	3350 (3100–3610)	<0.001

**Table 2 jpm-12-00762-t002:** Multivariate logistic regression model.

Variable	Odds Ratio (95% C.I.)	*p*-Value
rcs (EPW)	0.937 (0.928–0.947)	<0.001
rcs (EPW)’	1.067 (1.030–1.106)	<0.001
rcs (EPW)’’	0.813 (0.700–0.942)	0.006
Maternal age	1.050 (1.035–1.065)	<0.001
Maternal height	0.948 (0.937–0.959)	<0.001
Parity	0.639 (0.572–0.711)	<0.001
rcs (PAPP-A)	0.439 (0.031–0.591)	<0.001
rcs (PAPP-A)’	2.211 (1.490–3.066)	<0.001
β-HCG	0.880 (0.806–0.956)	0.004
Hypertension		
Chronic: no	2.887 (0.807–8.665)	0.075
Preeclampsia: no	4.885 (2.443–9.476)	<0.001
Gestational: no	3.854 (2.066–7.009)	<0.001
Smoking habits: no	0.479 (0.408–0.563)	<0.001
Abdominal circumference percentile	0.120 (0.066–0.217)	<0.001

**Table 3 jpm-12-00762-t003:** Screening performance for detection of small for gestational age (SGA) at birth.

Variable	AUC (95% C.I.)	Discrimination Rate (%) at 10% FPR
Abdominal circumference percentile	0.840 (0.829–0.850)	52.3
EPW	0.864 (0.854–0.873)	58.9
+Maternal age	0.865 (0.855–0.874)	59.4
+Maternal height	0.867 (0.859–0.878)	60.1
+Parity	0.873 (0.863–0.882)	60.7
+PAPP-A	0.874 (0.865–0.884)	61.5
+β-HCG	0.875 (0.865–0.884)	61.0
+Hypertension	0.877 (0.868–0.886)	61.8
+Smoking habit	0.880 (0.871–0.889)	61.8
+Abdominal circumference percentile	0.882 (0.873–0.891)	63.5

## Data Availability

The data analyzed were retrieved from the Miguel Servet University Hospital database.

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
