# Peer review of "Personalized Model to Predict Small for Gestational Age at Delivery Using Fetal Biometrics, Maternal Characteristics, and Pregnancy Biomarkers: A Retrospective Cohort Study of Births Assisted at a Spanish Hospital"

_jpm, 2022, doi:10.3390/jpm12050762_

Round 1

Reviewer 1 Report

Manuscript of Estaban is nicely written and methodology is correctly applied.  Results are interesting and improve our understanding in the field. However,  I have noticed that the introduction chapter needs significant improvement. In its current form it is very superficial and objective of study needs clarity and optimization. IMHO, study population should be mentioned in the title and elsewhere throughout the text.

As regards to English language: I suggest a thorough editing by a native speaker or professional service. 

Author Response

Reviewer 1

  • Moderate English changes required: Regarding the English language, we have requested an exhaustive review by a native speaker.

  • Does the introduction provide sufficient background and include all relevant references? : Can be improved

  • Are the conclusions supported by the results? : Can be improved

Manuscript of Estaban is nicely written and methodology is correctly applied.  Results are interesting and improve our understanding in the field. However, I have noticed that the introduction chapter needs significant improvement. In its current form it is very superficial and objective of study needs clarity and optimization. IMHO, study population should be mentioned in the title and elsewhere throughout the text.

Thank you for your suggestion. We considered to enlarge the title:  “Personalized model to predict small for gestational age at delivery using fetal biometrics, maternal characteristics, and pregnancy biomarkers. A retrospective cohort study of births assisted at a Spanish hospital between 2012 and 2018”.

We have modified the introduction and the objective as recommended. You can review it again in the manuscript.

We have completed conclusion:

Contingent screening models appear to be more sensitive than third-trimester ultra-sound screening as the sole technique for predicting SGA at delivery. However, these im-provements are modest (from 58.9% using EPW or 52.3% with AC percentile alone to 63.5% using the predictive model). AC at 35-week ultrasound does not appear to be supe-rior to EPW or significantly improve on the full model. 

In the other hand, study population is clearly specified in chapter 2 (Materials and Methods), 2.1 (study Design)

Reviewer 2 Report

The authors performed a novel and disruptive work on alternatives for a better assessment of occurrence of small-for-gestational-age newborns. The study itself and the results are well presented in the manuscript. I would recommend acceptance in current form.

Author Response

Reviewer 2

  • English language and style are fine/minor spell check required.

The authors performed a novel and disruptive work on alternatives for a better assessment of occurrence of small-for-gestational-age newborns. The study itself and the results are well presented in the manuscript. I would recommend acceptance in current form.

Thank you for your kind comment. We have reviewed English language and style by a native colleague.

Reviewer 3 Report

Major concerns

  1. This is an interesting manuscript but the methodology is suspicious  since fetal weight estimation is mainly by ultrasound, seriously follow up for high risk case for IUGR is the best way to find out targets. Clinically a suspected ultrasound at GA 35 weeks made the obstetrician to close monitor the fetal condition by ultrasound more than biometry but also  UA Doppler and AFI.
  2. . From the ultrasound exam to delivery there was approximately 30 days interval in the authors’ series, the growth condition may differ according to many conditions. Repeat ultrasound at 39 weeks may be more practical.

minor concern

The p value for "In vitro fertilization" in table 1 is not correct

Author Response

Reviewer 3

  • English language and style are fine/minor spell check required: We have reviewed English language and style by a native colleague.

  • Is the research design appropriate?: Can be improved
  • Are the results clearly presented?: Can be improved

  1. This is an interesting manuscript but the methodology is suspicious since fetal weight estimation is mainly by ultrasound, seriously follow up for high risk case for IUGR is the best way to find out targets. Clinically a suspected ultrasound at GA 35 weeks made the obstetrician to close monitor the fetal condition by ultrasound more than biometry but also UA Doppler and AFI.

  1. From the ultrasound exam to delivery there was approximately 30 days interval in the authors’ series, the growth condition may differ according to many conditions. Repeat ultrasound at 39 weeks may be more practical.

Thank you for the comments, we agree with the reviewer that best follow up for high risk cases of IUGR includes Doppler and amniotic fluid valuation. That’s what we do in those cases but not routinely in universal ultrasound screening at 35 – 37 weeks when estimated fetal weight is over 10th percentile. In this study we are only taking into account parameters that are performed routinely in low-risk pregnancies

Regarding your suggestion of repeated ultrasound at 39 weeks, it is a point to take into account when weighing the efficiency of the resources. As we considered in chapter 4 (Discussion): Detection of SGA at delivery by third-trimester ultrasound, even with models combined with other maternal variables and first-trimester biochemical markers, is limited, and new tools are needed. In this study we do not discuss the ideal time to perform the third-trimester ultrasound or if it is necessary to repeat it throughout the same one, but rather we evaluate whether a model that combines ultrasound performed universally around 35-37 weeks of gestation with other parameters improve de detection rate of SGA neonates.

Although a high predictive capacity is assumed if an ultrasound is performed at 39 week of gestation, the room for maneuver is limited, due to the proximity to delivery.

Minor concern: The p value for "In vitro fertilization" in table 1 is not correct

We thank the reviewer for his/her carefully review of the manuscript, the p-value was 0.394, we have corrected the mistake.

Round 2

Reviewer 3 Report

I have no further questions.